# CD8 Encephalitis in HIV: A Review of This Emerging Entity

**DOI:** 10.3390/jcm12030770

**Published:** 2023-01-18

**Authors:** Aniruddh Shenoy, Pavan Kaur Marwaha, Dominic Adam Worku

**Affiliations:** 1Haematology, Christie Hospital, Manchester M20 4BX, UK; 2Faculty of Medicine, University of Southampton, Southampton SO17 1BJ, UK; 3Infectious Diseases, Morriston Hospital, Swansea SA6 6NL, UK; 4Public Health Wales, Cardiff CF10 4BZ, UK

**Keywords:** CD8 encephalitis, HIV, central nervous system, neurology, viral escape

## Abstract

Introduction: Encephalitis is a life-threatening neurological condition with multiple causes in the setting of Human Immunodeficiency Virus (HIV). CD8 Encephalitis (CD8E) is a newly recognised condition which can present in an acute manner, with pertinent features including classical radiological findings with an intense brain parenchymal infiltration of CD8+ T cells. This review attempted to clarify the symptomatology, distribution and determinants of this condition, as well as to examine its vast unknowns. Methods: A literature review was undertaken in July 2022, utilising the PubMed and Google Scholar databases. Papers published between 2006–2022 were reviewed. Eighteen papers, totalling 57 patients, were found and analysed. Statistical analysis was undertaken using Chi-squared and Wilcoxon rank-sum tests as appropriate, with *p* < 0.05 deemed significant. Results: In this review, 57 patients were identified, with a female (61%, 34/56) and Black African (70%, 40/57) preponderance. Females were more likely to present with headache (*p* = 0.006), and headache was more likely to be present in those who died (*p* = 0.02). There was no statistically significant association between baseline CD4 count (*p* = 0.079) and viral load (*p* = 0.72) with disease outcome. Overall, 77% (41/53) of patients had classical imaging findings, including bilateral gadolinium-enhancing punctate and perivascular white matter lesions. However, many patients (23/57) required a brain biopsy as part of their diagnostic workup. Corticosteroid treatment was commonly prescribed in patients (64%, 35/55) and had a mortality benefit, with an overall survival in this group of 71% (*p* = 0.0008). In those who died, median survival was 5.5 months. In rare instances, recurrence of the disease was noted, which responded poorly to treatment. Discussion: CD8E represents a new and complex condition with few risk factors identified for its occurrence. The presenting symptoms are broad, but headache appears to be more common in females and more significantly associated with death. Though rare, CD8E is likely under-diagnosed, possibly due to overlapping features with other illnesses and lack of physician experience in its recognition and management. Corticosteroids demonstrate a clear mortality benefit, but more studies are required to determine their optimal dosing and duration, as well as the use of steroid-sparing agents. Further reviews should help to better determine the risk factors for the condition, as well as non-invasive biomarkers, to aid in diagnosis and help to predict poor prognosis and disease recurrence.

## 1. Introduction

Human Immunodeficiency Virus (HIV) is the virus responsible for Acquired Immunodeficiency Syndrome (AIDS), and despite concerted efforts, it remains a global health issue, with ~37 million people infected worldwide. Despite major advances in the diagnosis and management of HIV, it remains a leader in morbidity and mortality, particularly in Sub-Saharan Africa, where the majority of infections reside [1]. During the course of HIV infection, multiple cells of the immune system, including CD4+ T cells, dendritic cells and macrophages, are affected, leaving the individual susceptible to numerous opportunistic infections (OI) and non-infectious complications [2]. Furthermore, HIV may form latent reservoirs of infection within the kidney and central nervous system (CNS). These can persist even with treatment and lead to direct organ injury [3]. With the advent of antiretroviral therapy (ART), the landscape and prognosis of HIV infection has dramatically changed. The life expectancy of HIV-positive patients in the UK has increased by over a decade between 1996–2008 and is slowly approaching that of the general population [4].

Disorders of the CNS associated with HIV represent a complex facet of HIV disease management, with the aetiology depending on the degree of immunosuppression, stage of disease and treatment history [5]. They may present in a subacute manner, e.g., cryptococcal meningitis with signs of markedly increased intracranial pressure, focal deficits and seizures; or instead chronically, such as primary CNS lymphoma (PCNSL), in which cognitive decline, dementia and psychiatric disorders predominate [6,7]. Since the introduction of antiretroviral therapy (ART) and the routine use of chemoprophylaxis in at-risk individuals, there has been a marked reduction in CNS infection rates. However, this has not been paralleled with a decrease in non-infectious CNS complications, such as HIV-associated neurological disease (HAND)/HIV encephalitis. HAND represents a spectrum of neurocognitive defects that may affect up to 40% of ART-treated HIV patients, ranging from asymptomatic cognitive impairment to frank dementia [8].

Encephalitis is defined as the inflammation of the brain parenchyma, leading to neurological dysfunction that manifests as changes in cognition and consciousness. Though viral causes predominate, bacterial, autoimmune and fungal aetiologies can cause encephalitis in both immunocompetent and immunosuppressed patients [9]. CD8+ T cell encephalitis (CD8E) is a newly described condition, which is increasingly being recognised as a cause of encephalitis that is uniquely seen in the setting of HIV. It is characterised by substantial cerebral white matter CD8+ T-cell infiltration by an undefined mechanism and typical imaging findings on magnetic resonance imaging (MRI) [10]. First identified as a distinct entity in 2004, numerous further case reports have been documented in the literature, often retrospectively diagnosed by post-mortem autopsy. Contributing to this delay in diagnosis is the lack of understanding of its presentation and predisposing risk factors. Moreover, there is an overreliance on histological examination to confirm the diagnosis in individuals with no clear clinical characteristics, due to a lack of available surrogate serum biomarkers [10,11]. Given the plethora of CNS diseases in HIV, it is important for physicians to be aware of this diagnosis and how it presents, as well as to discriminate it from other, more commonly diagnosed conditions in this setting. In this review, we will explore the cases of CD8E reported in the literature to date and identify pertinent risk factors for the disease alongside common clinical manifestations and biochemical findings. Following this, management will be discussed alongside the relative unknowns of this condition.

## 2. Methods

A literature review was conducted for this narrative review in July 2022. Papers were identified by utilising the PubMed database and attempted to examine CD8E concerning its aetiology, presentation and treatment in HIV-positive populations. Papers were selected if they were written in English, published between 2006–2022 and had undergone peer review.

Search terms included ‘CD8+ T cells’, ‘HIV’ and ‘Encephalitis’. These terms were used in combination to identify search results. All authors independently participated in the evaluation of identified abstracts to ensure they fulfilled the study aims. All paper types, including reviews and case series/reports, were included in this analysis. Papers were excluded from analysis if information was inadequate, duplicated or incomplete, or if there were concerns of conflicts of interest or selection bias. From this initial search, 14 papers were found that met our inclusion criteria. A second literature search was undertaken in September 2022 including a screening of references, identifying an additional 5 papers. All selected papers were fully extracted and analysed. Data regarding patient demographics, symptomology, radiological and histological characteristics (where brain biopsy had been performed), cerebrospinal fluid (CSF) findings, treatment strategies, long-term outcome and preceding HIV control (e.g., viral load) were obtained where available and collated for analysis. In cases in which data were incomplete or not available, they were excluded from a given variable analysis. In cases of recurrence, data pertaining to the first incidence of CD8E were collected. A patient death was attributed to CD8E if either explicitly concluded so by the author or if clearly occurring during their presentation with CD8E.

For independent comparisons, *p*-values < 0.05 were considered significant. For comparison of group differences, the Chi-squared test and Wilcoxon rank-sum test were used where appropriate.

## 3. Results

We identified 19 papers in this review, which consisted of 57 patients (Table 1). Of identified cases in which gender was available, there was a clear female preponderance (61%, 34/56), with a median age of 42 years old (range: 19–69 years). Where ethnicity data were available, 70% (40/57) of patients were Black African.

Though only a minority of patients (3/57) presented on admission with CD8E as a presenting illness of undiagnosed HIV, 79% (45/57) of cases were on an established ART regime (range 0–17 years). Of the remaining cases (21%, 12/57), five had recently stopped therapy due to poor compliance and patient choice up to 5 months prior to their presentation.

The presenting features of patients were variable, with onsets over days to months. Common symptoms at presentation included headache, seizure and confusion; however, these occurred at different rates between males and females. In females, the most common symptom was headache, occurring in 62% (21/34) of females and only 27% (6/22) of males, respectively (Figure 1) (*p* = 0.006). Cognitive decline was the most commonly reported symptom in male patients, ranging from confusion to coma, being reported in 41% (9/22) of cases versus 12% (4/34) of females (*p* = 0.99). Overall features of vomiting and focal neurology were rarely reported. Importantly, constitutional symptoms were not described in any of the identified cases. Of those patients who died, headache was the most predictive, present in 64% (16/25) of patients (*p* = 0.02).

Of those with baseline serum CD4 performed at presentation (50/57), the median CD4 count was 315 cells/µL (range: 64–876 cells/µL), with a median serum HIV viral load (49/57) of 600copies/mL (range: 0–609,000 copies/mL).

Males (20/56) had higher baseline CD4 counts (median: 315 cells/µL) than females (median: 238 cells/µL). Baseline viral load was on average lower, but with a greater range in females (median: 373 copies/mL, range: 0–609,000 copies/mL) compared to males (median: 630 copies/mL, range: 0–67,976 copies/mL).

Only 18 patients had a peripheral CD8 count measured at presentation, with a median value of 888 cells/µL (range: 360–2001 cells/µL). When considering serum investigations where available by outcome (47/57), those who died (20/47) had higher peripheral viral loads (median: 1676 copies/mL) versus those who survived (median: 660 copies/mL) (*p* = 0.72). Patients that died demonstrated higher CD4 counts (median: 320 cells/µL) than those who survived (median: 227 cells/µL) (*p* = 0.079) (Figure 2).

CD4+ nadir was only reported in 20 cases, with a median value of 23 cells/µL (range: 5–516 cells/µL). Only three cases reported a CD4 nadir >200 cells/µL.

CSF analysis was available in 74% of patients (42/57). The median CSF white cell count was 29 cells/µL (range: 1–220 cells/µL) and was typically lymphocyte predominant. In cases in which the CSF lymphocyte count was subtyped, 90% demonstrated predominantly CD8+ cells with inverted CD4:CD8 ratio (18/20). The median CSF protein was moderately elevated at 0.92 g/L (range: 0.4–2.19 g/L). CSF HIV viral load was seldom performed (22/57), with a median value of 1200 copies/mL (range: 0–36,242 copies/mL), and it was often in excess of the paired peripheral HIV viral load. Those who survived had higher markers of neuroinflammation versus those who died, as indicated by a higher median CSF cell count (33 cells/µL versus 26 cells/µL, *p* = 0.947). However, in comparison to those who died, survivors demonstrated lower median protein levels (0.93 g/L versus 1.12 g/L) and CSF viral load (1100 copies/mL versus 4968 copies/mL) (Table 2).

In the 18 patients in whom a simultaneous CSF viral load and serum viral load assessment was made, there was a trend towards higher CSF viral loads in peripherally suppressed patients (Figure 3).

Of those with available imaging, 77% patients (41/53) had typical findings on MRI imaging, including bilateral, confluent and symmetrical high-signal-intensity lesions that were gadolinium contrast enhancing throughout the white matter and sometimes the grey matter, with preferentially perivascular distribution. Cases reporting only bilateral white lateral lesions without more specific features were labelled as semi-typical in nature and represented less than a quarter of cases (12/53). Overall, 40% (23/57) of patients received a brain biopsy as part of their diagnostic workup.

In 64% of patients (35/55), corticosteroids were used as part of a treatment regime. In those who received corticosteroids, the survival rate was 71% (25/35) (*p* = 0.0008) with treatment length varying considerably between 1–30 months. In those who did not receive steroids, death occurred in 60% cases. The overall case fatality rate was 45% (25/55). Relapse rate was low (*n* = 4), but occurred multiple times in one instance. Time to death was rarely explicitly reported (10/25), but varied from 2 days to 5 years post diagnosis, with a median survival time of 5.5 months. Overall follow-up time (21/35) ranged from between 3 months–8 years, with a median of 10 months’ follow up. Summary characteristics are described in Table 3.

## 4. Discussion

Encephalitis is a life-threatening CNS disorder characterised by inflammation of the brain parenchyma, and it can have various infectious and non-infectious triggers [31]. In the setting of HIV, the aetiology of CNS disease and encephalitis can stem from the direct HIV neurotropism with resultant secondary neuroinflammation, as evidenced in the 20–50% of HIV patients suffering with HAND [32,33]. Other causes include secondary opportunistic infection (e.g., bacterial, parasitic, fungal) [34], neurotoxicity from ART itself and Immune Reconstitution Inflammatory Syndrome (IRIS) [35,36]. Thus, careful consideration and multimodal investigation are required to delineate the cause.

In 2004, Miller et al. documented the first case of CD8 encephalitis (CD8E) [11]. Two cases were described of acute encephalopathy in HIV-positive patients who had marked cerebral inflammation with a significant presence of CD8+ T cells on post-mortem histology. From this initial description, a stream of similar cases has been retrospectively recognised, often on post-mortem analysis. As such, CD8E is now recognised as a distinct neurological disease. The phenomenon of diffuse perivascular and parenchymal infiltration of CD8+ T cells with associated microglial activation, reactive astrocytes, weak expression of HIV protein p24 in astrocytes and lack of giant multinuclear cells typifies and differentiates CD8E from other common CNS disorders [10,11].

As shown in our review, CD8E often presents in an acute/subacute manner with headache, confusion and seizures, and rarely with ataxia and drowsiness. Emergent findings in the demographics of our cases are the wide range of ages amongst cases (19–69 years) and clear female and Black African predominance, which has been commented upon previously [13]. Though this mirrors the global epidemiology of HIV infection, it may suggest a genetic predisposition or viral determinants that are more prevalent in Black African communities, which could facilitate CD8E occurrence and requires further investigation. This may also be supported by the different patterns of symptoms observed between males and females. Headache was significantly more common in females (*p* = 0.006), whereas cognitive decline was more common in males. The symptomatology of CD8E, however, is markedly differentiated from HAND, in which a spectrum of cognitive impairment, including a dementia-like process with marked changes in executive function and impulse control, is seen. Typically, this occurs in elderly patients in a chronic manner, even amongst those with good viral control [37,38].

As our data show, CD8E can occur within both ART-naïve and well-controlled HIV patients, with no clear preponderance to treatment status or degree of immunosuppression (CD4 count range: 64–876 c/µL). Though the majority of our cohort was well established on ART at the time of presentation, three patients presented as a new diagnosis of HIV with CD8E. This would suggest that CD8E should be considered an indicator condition for HIV testing, as it has not been described in any other immunocompromised population group to date. Moreover, this makes the risk of an iatrogenic pathophysiology underlying CD8E somewhat less likely.

In the 35% (20/57) of our cohort with available CD4+ nadir measurements, the average value was 96 cells/µL. A lower CD4+ nadir has been observed to be a predictor of neurocognitive impairment and HAND, and it has been linked to the establishment of a viral reservoir in the CNS and subsequent CSF escape [39,40]. Accordingly, a low CD4+ nadir, and hence late HIV diagnoses, may be risk factors for the incidence of CD8E. A retrospective analysis of 14 patients by Lescure et al. does not indicate any obvious relationship between CD4+ nadir and neurological recovery, nor do our collated data suggest any relation to mortality [18].

Lumbar puncture (LP) is an essential investigation in the diagnosis and management of encephalitis in HIV and non-HIV patients. Rather strikingly, CSF analysis data were not available in a quarter of cases (15/57). The most common reason for not performing an LP was the concern or presence of symptoms of raised intracranial pressure (ICP). This is a commonly cited reason for not performing an LP, given the risks of cerebral herniation. However, LP should be performed in all patients upon ruling out raised ICP on neuroimaging, as it is critical in establishing aetiology by means of serology, culture and molecular methods [31]. In fact, our paper finds only the presence of headache to be a significantly predictive factor for mortality in CD8E (*p* = 0.02), which, if linked with increased ICP, would explain the paucity of CSF investigations undertaken in our cohort. Electroencephalogram (EEG) studies were rarely reported, but often showed diffuse and generalised slowing, with no signs of epileptiform activity. Though these are non-specific findings, it is important to note that the value of EEGs is in diagnosing conditions mimicking encephalitis such as non-convulsive status epilepticus. They can also provide aetiological clues, such as in HSV encephalitis, and so should be encouraged [9].

Though rarely performed in conjunction with peripheral viral load (*n* = 21), our data show that CSF viral load can be highly raised (median 1200 copies/mL), even in the setting of a suppressed plasma viral load. The significance of CSF viral load is important, given its correlation with neurological dysfunction, although the significance of low-level CSF viraemia is uncertain in patients on existing ART [41].

Several cases have reported a novel HIV viral mutation(s) found at the time of symptoms within the CNS and not in the periphery. In conjunction with the raised CSF viral load, this is highly suggestive of viral escape, which may be a trigger for CD8E development [13,19]. Indeed, in the case presented by Morioka et al., the patient was treated only by altering ART regime with no additional treatment, suggesting this may be a treatment option in select patients [15]. Regardless, it may be useful to consider genotypic resistance testing in all patients already on ART who present with CNS disease, particularly in the setting of possible CD8E.

CSF analysis in CD8E clearly demonstrates a marked lymphocytic pleocytosis (29 cells/mL) alongside a significantly raised protein (0.92 g/L) in the setting of normal CSF: plasma glucose ratio [13,14,15,16,17,18,19,20]. Higher CSF white cell counts were recorded in those who survived, though there was no statistically significant relationship, and this may reflect a more qualitative and quantitatively intact immune system (*p* = 0.947). Where CSF lymphocyte subsets were tested, 90% demonstrated predominantly CD8+ lymphocytes, often with a significant CSF CD8+/CD4+ ratio up to 5:1 in nature [14,16,17,18,19,22,23,30]. Though increased CSF CD4:CD8 T-cell ratios have been noted in several conditions, including neurosarcoidosis [42] and myasthenia gravis [43], the causes of a raised CSF CD8:CD4 T cell ratio have rarely been described outside of HIV CSF viral escape and neuro-IRIS, with the latter of these particularly seen during ART initiation [42]. This suggests that a disproportionate abundance of CSF CD8+ T cells is a prevailing feature of CD8E and may be used as an early diagnostic indicator of the disease. This further reinforces the value of CSF analysis in diagnosing encephalopathies in HIV, given that this is a condition in which early recognition has been demonstrated to result in better clinical outcomes.

Of interest, in one case by Kerr et al. [19], low-level CSF EBV DNA positivity was found with the presence of EBV-encoded small RNAs (EBER) on brain biopsy and histology. Though EBV is likely responsible for 2–5% of viral encephalitis cases, the significance of this result is unknown [44]. EBV seropositivity amongst adults is 90%, with the majority of patients having asymptomatic infection or glandular fever as a young adult [45]. Importantly, EBV, like other herpes viruses, can reactivate whereby it can lead to dangerous complications in the immunocompromised host, including post-transplant lymphoproliferative disease (PTLD). Therefore, in the setting of immunosuppression, it is not uncommon to find EBV reactivation, which can occur in the CSF with the presence of co-existent CNS infection, such as tuberculosis, toxoplasma, aspergillus and herpes simplex virus [46]. Therefore, it is unknown whether positive EBV PCR in the arena of CD8E represents cause or effect, but this is worthy of further research.

A notable observation is the high rate of brain biopsy as part of the diagnosis and management of patients with CD8E. In our case series, 23 (40%) cases had biopsy undertaken as part of their initial management. Though this may seem appreciably high, it is important to consider that in retrospective cohort studies (*n* = 340), 57% of confirmed encephalitis cases had no cause identified [47]. As such, invasive brain biopsy remains a useful and likely under-utilised tool in those for whom a specific diagnosis cannot be made by conventional means. Though routine use of brain biopsy is not warranted in the majority of patients with CNS disease, it remains the gold standard in encephalitis and in those patients with HIV presenting with mass lesions. Importantly, brain biopsy has a sensitivity of up to 96% depending on the reporting neuropathologist, with evidence in one review of biopsy in the setting of HIV to have changed management in 57.7% of patients [48,49]. Though very invasive, it is important to note there is convincing safety data of brain biopsy in the setting of CNS disease, with a mortality and morbidity of 5.7% and 0.9% respectively [48,49]. Though often impractical given the deep location of lesions in certain conditions, in CD8E, given the propensity for surface structures, it remains a viable means to diagnose this challenging condition by means of intense CD8+ T cell infiltration and to exclude mimics, providing contraindications do not exist.

Radiological imaging is a key tool with which to diagnose encephalitis, with classical findings being well described in the literature for certain aetiologies, such as temporal lobe hyperintensities in herpes simplex virus (HSV) encephalitis [50]. Therefore, when available, MRI represents the most sensitive and specific neuroimaging technique for the evaluation of encephalitic patients, with diffusion-weighted imaging offering superior diagnostic rates versus conventional MRI [50]. In our analysis, all patients underwent neuroimaging with MRI, with 77% (41/53) of cases demonstrating typical imaging findings. These findings, which now characterise CD8E, include diffuse bilateral white matter signal changes described as punctate or linear and perivascular in nature, which is unique and often associated with post-gadolinium enhancement on T1 sequencing (Figure 4). Other less frequently described findings include grey matter changes, generalised sulcal effacement and changes to vascular integrity [13,20,21]. Rare presentations have been described with CD8E mimicking posterior reversible encephalopathy syndrome (PRES), solid tumour and autoimmune encephalitis [17,20,30].

Periventricular and deep white matter hyperintensities are also seen in HAND, which may be enhanced upon contrast imaging [51]. However, it is uncommon to see any changes in early disease, and these appear to be far less prominent than in CD8E, but may more commonly affect the deep structures of the brain on T2-weighted and fluid-attenuated inversion recovery (FLAIR) sequences.

Other common diagnoses to consider in the setting of predominant white matter lesions of the CNS in the setting of HIV include PCNSL, Progressive Multifocal Leukoencephalopathy (PML) and Acute Disseminated Encephalomyelitis (ADEM). PCNSL is a non-Hodgkin lymphoma that is intricately related to EBV infection, an important consideration in severely immunosuppressed patients, and it occurs in 2–6% of HIV patients with advanced disease (CD4 < 100 cells/uL). However, it classically presents with rapidly acute symptoms of personality change and aggression, which are commonly described, but no seizure or signs of increased ICP, such as in CD8E, with imaging often showing single/multiple focal lesions with central necrosis and haemorrhage [52].

PML, caused by reactivated JC virus, is linked to secondary immunosuppression from medications (e.g., Natalizumab), but is most commonly described in the setting of HIV infection, with an incidence of 0.7 per person-year of follow up [53]. Classically, symptoms worsen in a chronic fashion with seizures occurring in a subset. On MRI T2 Flair sequences, hyperintense demyelinating lesions are seen, which are hypointense on T1 sequences, helping to differentiate it from HAND and CD8E [54].

ADEM, like PML, is typified by demyelination throughout the CNS, but it ordinarily has a trigger, which may include preceding viral infection or vaccination receipt, but it can be secondary to HIV infection. Unlike in CD8E, typical features include predominant motor dysfunction and ataxia with concurrent fever and seizures. Like CD8E, symptoms can be subacute, although hyperacute forms complicated by haemorrhage are recognised [55,56]. MRI in this setting shows multiple lesions in the grey and white matter on T2-weighted sequences, with variable gadolinium uptake making it indistinguishable from multiple sclerosis [57].

As such, amalgamating radiological, clinical and biochemical correlates can help the attending physician in the recognition of CD8E.

Across reported cases in the literature, it appears that CD8E is a steroid-responsive disease, with marked improvements in neurological deficits and radiological findings seen amongst recipients [13,18]. In our review, 71% patients (25/35) who received corticosteroids survived, versus a 60% death rate in those who did not receive them (15/25). This is an appreciably higher survival rate than toxoplasma encephalitis and cryptococcal meningitis, which are amongst the most common CNS disorders in HIV, with survival rates of ~30% and 40% respectively [58,59]. Indeed, the use of steroids was statistically significantly linked with survival (*p* = 0.0008), making this a useful intervention in suspected cases. Commonly employed regimens have involved intravenous methylprednisolone pulsed over 3–5 days, followed by an oral steroid taper over durations ranging from 1 month to 30 months, most commonly with prednisolone. Importantly, there was variability in the times until treatment initiation, which was up to 33 days in one case [18].

Published cases show varying recovery timeframes, which range from a few days to several months, with as of yet no discriminating factors for responders versus non-responders. Although follow up was performed in most survivors, it was done so for a relatively short period of time (median of 10 months). As such, we presently cannot be sure of any long-term consequences of CD8E in the follow up of survivors [18,19].

Given the plethora of side effects from chronic steroid use, the use of a steroid-sparing regime is an important consideration, as is the general use of immunosuppression in already immunosuppressed individuals. Mycophenolate Mofetil (MMF) has been employed for this purpose and seems to be a good choice given its relatively good CSF penetration and tolerability. In three reviewed case reports of its use, it allowed for sustained improvements in neurology and steroid dose weaning, but it did not necessarily alter the outcome, with two of these cases dying [21,22,30].

Of note, MMF has been used in two cases of relapsing CD8+ with contrasting outcomes. Salam et al. reported success in employing MMF as a means of offsetting steroid dependency in a case of multiple recurrences of symptomology [21]. However, Thom et al. reported on a patient who initiated MMF therapy as a salvage therapy after relapsing four months following intravenous methylprednisolone treatment for CD8E. Subsequent radiological imaging demonstrated progressive changes at 1 year, with the patient eventually succumbing to sepsis [22]. As such, the efficacy of MMF as a steroid-sparing agent remains to be established, though it remains the only agent found in the literature to be trialled for this purpose.

CD8E is very rarely observed to follow a polyphasic course following treatment, though Salam et al. reported multiple relapses over a 3-year period, directly relating to steroid weaning. Interestingly, in two papers reviewed, CSF viral load was seen to rise between recurrences despite prior treatment and alongside stable peripheral viral loads, raising the question as to whether CSF viral escape may be the driving pathophysiology behind relapsing disease and could be a marker to predict recurrence [21,22].

## 5. Conclusions

CD8E represents a new and poorly understood condition which has the potential to affect HIV patients independent of viral control. Though typically associated with abrupt symptoms, it remains clear that it can lead to protracted disease with a high case fatality ratio and an overreliance on brain biopsy to diagnose. It is apparent from our review that more robust information is needed to understand the natural history of this condition. Certainly, wider use of CSF analysis, including CSF viral load and CD8+ T lymphocyte count, as well as a better understanding of patient characteristics will be needed to allow for advances to be made. Importantly, though few cases exist in the literature, it is likely that many of the undifferentiated encephalitis diagnoses found in clinical practice in HIV patients could be ascribed to this. Given the high fatality rate, CD8E should be considered in all HIV patients presenting with encephalitic disease, regardless of premorbid status, with the timely initiation of immunosuppressive therapy alongside a review of ART and genotypic resistance testing.

## Figures and Tables

**Figure 1 jcm-12-00770-f001:**
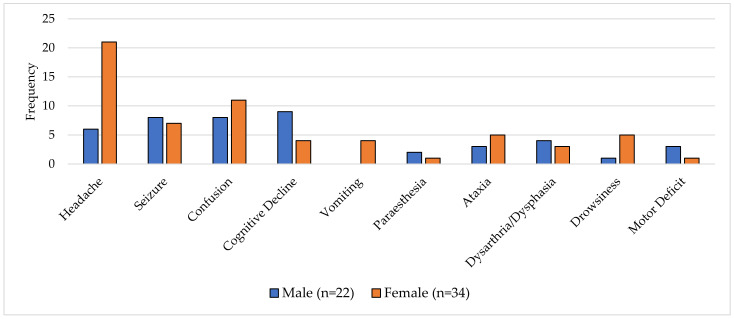
A graph of presenting symptoms of CD8E by sex (*n* = 56).

**Figure 2 jcm-12-00770-f002:**
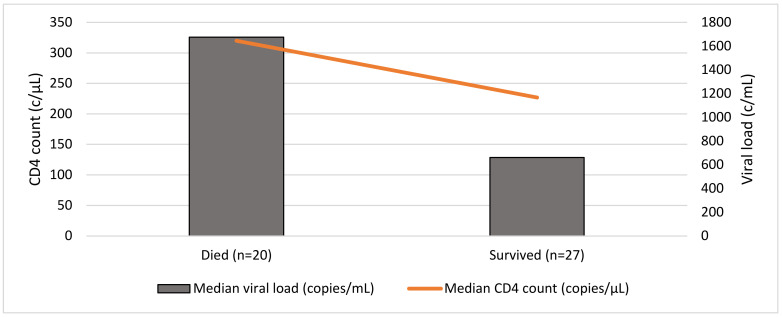
Baseline CD4 count and viral load at presentation and outcome (*n* = 47).

**Figure 3 jcm-12-00770-f003:**
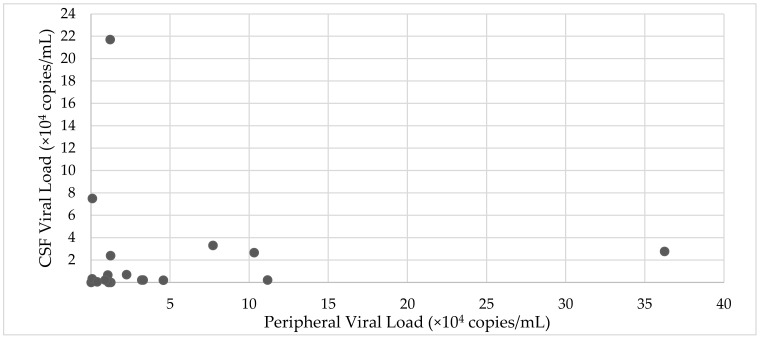
Relationship between CSF viral load and peripheral viral load (*n* = 21).

**Figure 4 jcm-12-00770-f004:**
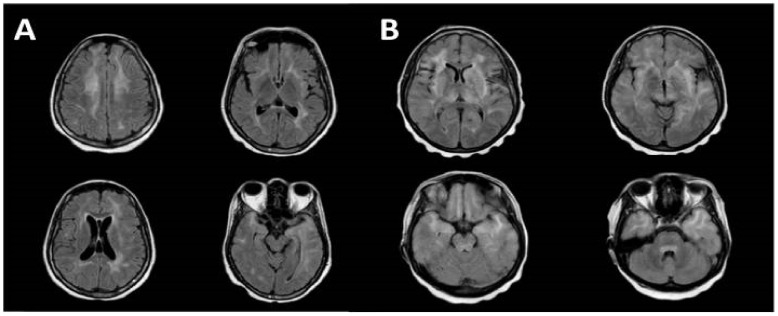
MRI brain images demonstrating the characteristic features in CD8E. (**A**) bilateral periventricular deep white matter changes with high signal involving both temporal lobes on axial T2 FLAIR sequences. (**B**) axial T2 FLAIR sequences revealing diffuse non-restricting T2 hyperintensities throughout the white matter involving the pons, temporal poles, insulae and periventricular white matter (with permission from Kerr et al. [19]).

**Table 1 jcm-12-00770-t001:** Demographics, baseline characteristics and outcomes of 57 patients with CD8+ Encephalitis [12,13,14,15,16,17,18,19,20,21,22,23,24,25,26,27,28,29,30].

Age (Yrs)	Gender	Serum Analysis			Imaging	CSF Analysis			Died	Brain Biopsy	On ART	EEG Findings
		CD4+ (c/µL)	CD8+ (c/µL)	pVL (c/mL)		Cells (c/µL)	Protein (g/L)	VL (c/mL)				
32	F	95	NA	291	Typical	12	0.4	NA	Yes	No	Stopped	NA
40	M	543	NA	21,359	Typical	17	1.35	NA	Yes	No	No	NA
28	F	876	2001	<50	Typical	20	NA	NA	Yes	No	Yes	NA
46	F	170	960	86,800	NA	NA	NA	NA	Yes	No	Stopped	NA
40	M	315	306	<50	Typical	NA	NA	NA	Yes	No	Yes	NA
36	M	>400	NA	3568	Typical	NA	NA	NA	Yes	No	Yes	NA
43	F	240	1290	<50	Typical	8	0.68	NA	Yes	No	Yes	NA
47	F	824	1755	238	Typical	NA	>0.8	NA	Yes	No	Yes	NA
49	F	374	970	12,062	Typical	NA	NA	NA	No	Yes	Yes	NA
29	F	560	960	<50	Typical	NA	NA	NA	Yes	No	Yes	NA
41	F	266	NA	439	NA	NA	NA	NA	No	No	Yes	NA
44	M	233	896	<50	Typical	80	1.2	<50	Yes	No	Yes	NA
37	F	353	880	8759	Typical	NA	NA	NA	Yes	No	Stopped	NA
19	M	64	NA	600	Typical	NA	NA	NA	No	Yes	No	NA
33	F	200	1340	8300	NA	NA	NA	NA	Yes	No	Yes	NA
51	F	NA	NA	NA	Typical	NA	NA	NA	Yes	No	Yes	NA
52	F	220	360	<50	Typical	NA	NA	1100	No	Yes	Yes	NA
33	F	384	741	3300	Typical	NA	NA	7700	Yes	No	Yes	NA
35	F	487	703	125,893	Typical	NA	NA	NA	No	Yes	Stopped	NA
52	F	NA	NA	NA	Typical	NA	NA	NA	Yes	No	Yes	NA
45	M	320	NA	4000	Typical	23	1.15	NA	Yes	No	Yes	NA
45	M	NA	NA	NA	NA	NA	NA	NA	Yes	No	Yes	NA
69	M	460	820	<50	Typical	26	0.56	<50	Yes	No	Yes	NA
54	F	336	NA	16,700	Typical	45	0.93	NA	No	Yes	No	NA
58	M	NA	NA	NA	Typical	2	0.76	NA	No	Yes	Yes	NA
52	M	632	NA	660	Semi- Typical	18	1.29	1050	No	No	Yes	NA
27	F	103	782	<20	Typical	NA	NA	NA	No	Yes	Yes	NA
46	M	121	NA	4500	Typical	100	1.47	NA	Yes	Yes	Yes	NA
41	M	120	NA	35,561	Typical	9	1.1	NA	No	Yes	Stopped	NA
36	M	93	NA	0	Typical	40	0.63	0	No	Yes	Yes	NA
47	F	275	NA	692	Typical	80	0.9	2236	Yes	Yes	Yes	NA
39	F	NA	NA	NA	Typical	26	0.9	1120	Yes	Yes	Yes	NA
33	F	283	NA	2660	Typical	20	1.13	10,300	Yes	Yes	Yes	NA
37	F	495	NA	65,800	Typical	220	0.79	NA	No	Yes	Yes	NA
54	F	402	NA	NA	Typical	220	0.8	672	No	Yes	No	NA
33	M	210	NA	2379	Typical	1	0.42	1230	No	Yes	Yes	NA
43	M	84	NA	2765	Typical	46	1.1	36,242	No	Yes	Yes	NA
35	M	214	NA	21,700	Typical	80	0.8	1200	No	No	Yes	NA
59	F	NA	NA	NA	Typical	30	0.52	NA	Yes	No	Yes	NA
49	M	114	NA	200	Typical	19	1.57	3200	No	No	Yes	NA
39	M	742	NA	201	Typical	26	1.1	3294	No	No	Yes	NA
52	F	409	1807	7500	Semi- Typical	200	0.62	72	No	Yes	No	NA
39	F	125	NA	<40	Semi- Typical	42	1.21	NA	No	Yes	Yes	Diffuse generalised slowing
33	F	71	NA	609,000	Semi- Typical	39	1.14	NA	No	Yes	No	None
42	F	234	NA	0	Semi- Typical	33	2.19	1240	No	No	Yes	Bilateral slowing
37	F	140	NA	330,000	Typical	NA	NA	NA	No	Yes	No	None
34	F	NA	NA	NA	Semi- Typical	NA	NA	NA	No	No	Yes	No
52	F	220	360	0	Typical	5	0.44	1100	No	Yes	Yes	Generalised slowing, right frontal sharp waves
50	F	172	NA	308	Semi- Typical	NA	NA	<60	No	No	Yes	Nil seen
55	M	525	NA	200	Typical	30	0.91	11,157	No	No	Yes	No
51	M	588	NA	<50	Semi- Typical	6	0.89	378	NA	No	Yes	No
49	M	520	NA	184	Semi- Typical	14	0.74	4570	NA	No	Yes	No
35	F	397	546	228	Semi- Typical	33	1.02	NA	No	No	Yes	Diffuse theta-range slowing
56	F	474	NA	<200	Typical	7	0.73	886	No	No	Yes	Nil
27	M	340	NA	67,976	Semi- Typical	29	1.33	NA	No	No	Yes	No
37	F	610	NA	120	Typical	36	1.12	NA	Yes	No	Yes	No
43	NA	444	NA	8215	Semi- Typical	55	1.79	NA	Yes	Yes	Yes	No

Abbreviations: NA—not available; pVL—peripheral viral load; VL—viral load; ART—antiretroviral therapy; EEG—electroencephalogram.

**Table 2 jcm-12-00770-t002:** Cerebrospinal fluid analysis (median values) by CD8E outcome.

Outcome	CSF Cells (c/µL)	CSF Protein (g/L)	CSF VL (c/mL)
Died (*n* = 25)	26	1.12	4968
Survived (*n* = 30)	33	0.93	1100

Abbreviations: CSF—cerebrospinal fluid; VL—viral load.

**Table 3 jcm-12-00770-t003:** Summary characteristics of patients (*n* = 57).

Characteristics	Data
Sex—n Male Female	22 34
Age—years Median (interquartile range) Range	42 (35–50) 19–69
On ART at presentation—n (%)	45 (79)
Serum investigations (n) Median peripheral viral load—copies/mL Median presenting CD4+ cell count—cells/µL Median serum CD8+ cell count—cells/µL	600 (49) 315 (50) 888 (18)
Typical radiological imaging findings, where available—n (%) Brain biopsy—n (%)	41 (77) 23 (40)
Case fatality rate—n (%)	25 (45)
Receipt of Corticosteroid—n (%)	35 (64)
Median survival time—months (n) Median follow up—months (n)	5.5 (10) 10 (21)

Abbreviations: ART—antiretroviral therapy.

## Data Availability

Not applicable.

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
