# Peer review of "CD8 Encephalitis in HIV: A Review of This Emerging Entity"

_jcm, 2023, doi:10.3390/jcm12030770_

Round 1

Reviewer 1 Report

In the manuscript, to clarify the symptomatology, distribution and determinants of CD8 Encephalitis as well as examine its vast unknowns, Dr. Shenoy and coauthors reviewed eighteen papers, analyzed the data from 57 patients in total and concluded that CD8E should be considered in all HIV patients presenting with encephalitic disease. I have multiple concerns as follows:

1.     In abstract, the authors stated “Results showed a female (65%)”. However, in Results section, it said “(Male: Female ratio 0.65)”. Please explain the inconsistency or correct the sentences appropriately.

2.     The statistical analysis section is missing in the Methods section. A detailed description about the analysis methodology is recommended.

3.     In general, it is recommended to present results with the percentage and numbers of patients in both numerator and denominator. For example, in line 96 “Where ethnicity data was available, 70% (n=40) of cases were Black African”, 70% (40/57) cases are Black African; in line 115 “Of those patients who died ... the presence of headache being present in 64% (n=16) of patients.”, 64% (16/25) of patients who died had headache. In addition, a flow chart or a summary table describing study population and subsets of interests (e.g. patients who died; patients with symptom ...) with specific sample size would be welcomed to avoid confusion for reader.

4.     Is there any formal testing for any comparison, such as whether there is difference in proportion of patients with cognitive decline between male vs. female; whether there is difference in proportion of patients with headache between patients who died or not?

5.     In Figure 1, please clarify that there are 46 patients with symptom to avoid any misunderstanding.

6.     The biomarkers of interest are not normal distributed. Besides mean and range, geometric means with CI could be summarized. Also, please specify how missing data was handled.

7.     In line 122 “Males (n=20) ...”, please provide the sample size for females as well.

8.     There is a table without any label, caption, or description between line 130 and 131.

9.     In figure 3, please clarify why only 46 patients were included in the analysis and provide the geometric means with CI in addition to the average values.

10.  In results section, a summary table for variables of interest (e.g. death, corticosteroids, ...) is recommended.

11.  In addition to the fatality rate, further information could be reported, such as follow-up time, median survival time estimates, cumulative incidence rate over time. Survival models may be fitted as appropriate.

12.  There are quite a bit grammatical errors and ambiguity throughout the manuscript, please try going through the manuscript and revise as much as possible.

Author Response

Dear Reviewer,

Thank you very much for your evaluation. Please find below our replies:

  1. We have corrected this value, (females 61%)
  2. We have included our statistical methodology (Chi-squared test and two-sided t-test, geometric mean with 95% CIs and p values)
  3. We have incorporated percentages and numbers of patients as described throughout the paper, as well as a table of subsets of interest with patient numbers
  4. We have used Chi-squared test and two-sided t-test where appropriate
  5.  Clarification made
  6. Geometric mean and CIs are now included. We have stated explicitly that missing data were excluded from a given variable analysis
  7. Sample sizes incorporated
  8. Tables and Figures now labelled
  9. As per point 6, missing data were excluded from a given variable analysis
  10. Table of subsets of interest with patient numbers included in results
  11. Follow up times and median survival times incorporated and discussed in discussion
  12. Paper re-reviewed for grammar by all authors

Reviewer 2 Report

The reviewers read with great interest the submitted article on CD8 encephalitis. The paper discusses papers reported in the past 2006-2022 based on 57 cases extracted using a search engine with keywords such as "HIV," "CD8 + TCell," and "encephalitis."

Below are several questions from the reviewers. I would appreciate the authors' opinions and comments on my questions.

Major1
Regarding the name of encephalitis, we believe that CD8 encephalitis is characteristic of HIV, but is it also seen in other immunodeficiency diseases in which CD4/CD8 is associated? Or have there been reports of such?

Major2
Please provide MRI images of CD8 encephalitis in this paper.

Major3
Do the MRI images of CD8 encephalitis have any specific imaging findings that differentiate them from the MRI images of the many other encephalitic encephalopathies or other HIV-associated encephalitis?

Major4.
What do you think of the possibility that CD8 encephalitis could be a drug-related encephalitis used as HIV, as other conditions are known to be drug-related encephalitis/encephalopathy associated with cyclosporine and chemotherapy?

Major5
The diagnosis of encephalitis usually requires a change in level of consciousness and electroencephalographic findings in addition to those listed in the table. Is it possible to add to the table the findings and data of the electroencephalography in particular?

Major6.
JCM is a general medical journal, so the definition of encephalitis should be properly stated in the Introduction.
The following references are provided for your reference.

A comprehensive review of pediatric acute encepahlopathy

JCM 2022 Oct 7;11(19):5921 doi: 10.3390/jcm11195921.

Best regards,

Dr. Reviewer

Author Response

Dear Reviewer

Thank you for your evaluation. Please find below our replied

Major 1 - We clarify that CD8E has only been reported in the context of HIV, but we discuss other instances where CD4/8 ratios are of interest

Major 2 - MRI images provided

Major 3 - The typical constellation of bilateral, confluent, and symmetrical high signal intensity lesions with gadolinium contrast enhancement bilaterally with preferential perivascular distribution is discussed though we acknowledge other conditions with elements of comparable MRI findings, and so emphasise the importance of the holistic approach to diagnosis

Major 4 - Given the range of treatment status and duration, from naive to long-term treatment as well as recently ceased, we argue that this would not be in keeping with an iatrogenic aetiology regarding HIV ART

Major 5 - We have incorporated EEG findings that were reported on

Major 6 - Thank you for the reference, with respect we have defined encephalitis referencing a source discussing encephalitis in adult patients

Round 2

Reviewer 1 Report

The authors addressed majority of my concerns. In addition, a revision is recommended to improve the quality of the article addressing concerns and questions listed below:

1.     T-test is not appropriate for variables not normal distributed. Wilcoxon rank-based test is recommended without normality assumption.

2.     In line 113 “there was a clear female preponderance (61%, n=34/56) ...”, it would be better to write as “(61%, n=34/56)” without “n=”, since it’s not “n” but a ratio between No. of female and No. of total subjects. Please try going through the manuscript and revise accordingly.

3.     In table 2, numbers presented for all measurements, CSF Cells, CSF Protein, CSF VL, are means among died/survived subjects? The unit for CSF Cells is missing, which should be (c/μL). Then, what doesn’t “n” means? Number of subjects with records available? It doesn’t make sense to include subjects with partial missing values (e.g. Cells and Protein record available but VL not available; Cells record available but VL and Protein not available) but exclude subjects without all three measurements. All subjects should be included in the table. Please consider reorganizing the table structure as example shown below.

1.     In figure 3, it could be better to report geometric mean, since both variables are not normal distributed. Or the authors could provide this information in main text without modifying the Figure. Wilcoxon test p value can be provided as well. In addition, Figure 2 is missing in current version of manuscript.

2.     Table 3 summarized important characteristics of patients, which is helpful for audiences. But there is too many information included. And mixed summaries (Numbers, ratio, p values) were presented without clear labels.  Please reorganize Table 3 to make it easier to read.

Author Response

16/12

Dear Reviewer,

Thank you very much for your evaluation. We are underway with amendments but on further interrogation of our statistics, we wondered if we could clarify a few things before our next submission

  • Regarding variables that are not normally distributed, we acknowledge that the arithmetic mean is affected by skew. We have encountered trouble in processing the geometric mean due the presence of zero values in multiple datasets. We find quite different values if we omit these zeroes (overestimating the average) or replace them with "1", and are not comfortable that either approach is appropriate. We put forward that summarising these variables with medians and ranges might be the most pragmatic option instead?
  • We are not sure if confidence interval calculations are appropriate for our data as they are a random sample of a larger population, nor are most variables normally distributed, both of which would generally be required for confidence interval calculations. As such, we ask whether if we can omit the confidence intervals?
  • Regarding your suggestion point 3, you refer to an example table template but we are not able to see it. Is it possible to be provided with this please?
  • To clarify, we are undertaking Wilcoxon rank-sum testing in place of T-testing as the relevant datasets are non-parametric and independent

18/12

Further to the above, Figure 3 has been modified to present median values instead of means

Titled has been slightly altered as per Reviewer 2's feedback

Many apologies, we were not able to anticipate the table structure for point 3 that you referred. We have substituted in median values but unfortunately have not changed the structure much beyond this. Table 1 summarises the patients for whom individual CSF parameters were available  

Reviewer 2 Report

The reviewers reviewed the manuscript of the article as amended by the authors. The absence of EEG findings is disappointing. However, the high quality MRI images are clinically meaningful. Two new literature citations have also been added, and the discussion is better organised in terms of differential diagnosis and argumentation. I also thought the tables and graphs were clearly organised.
The reviewers support the publication of this review in JCM.

One point: the title "Stage-1" should be deleted. The title is also attractive as it defines a new encephalopathy. But does it really reflect the content of the paper? I hope the reviewers will reconsider this title only from the author's point of view.

Best regards,

Dr. Reviewer

Author Response

Dear Reviewer,

Thank you very much for your feedback. Your advice regarding the title is duly noted, we have altered the title to now read "CD8 Encephalitis in HIV: A Review of This Emerging Entity

Changes have been made from reviewer 1's feedback so we reuploaded an updated draft